# Prediction and Visualization of Non-Enhancing Tumor in Glioblastoma via T1w/T2w-Ratio Map

**DOI:** 10.3390/brainsci12010099

**Published:** 2022-01-12

**Authors:** Shota Yamamoto, Takahiro Sanada, Mio Sakai, Atsuko Arisawa, Naoki Kagawa, Eku Shimosegawa, Katsuyuki Nakanishi, Yonehiro Kanemura, Manabu Kinoshita, Haruhiko Kishima

**Affiliations:** 1Department of Neurosurgery, Asahikawa Medical University, Asahikawa, Hokkaido 078-8510, Japan; shota.ikao@gmail.com (S.Y.); kyokui100084@gmail.com (T.S.); 2Department of Neurosurgery, Osaka University Graduate School of Medicine, Suita 565-0871, Japan; nkagawa@nsurg.med.osaka-u.ac.jp (N.K.); hkishima@nsurg.med.osaka-u.ac.jp (H.K.); 3Department of Diagnostic and Interventional Radiology, Osaka International Cancer Institute, Chuo-ku, Osaka 541-8567, Japan; xiaohumeixu@gmail.com (M.S.); je2k-nkns@asahi-net.or.jp (K.N.); 4Department of Diagnostic Radiology, Osaka University Graduate School of Medicine, Suita 565-0871, Japan; a-arisawa@radiol.med.osaka-u.ac.jp; 5Department of Molecular Imaging in Medicine, Osaka University Graduate School of Medicine, Suita 565-0871, Japan; eku@mi.med.osaka-u.ac.jp; 6Department of Biomedical Research and Innovation, Institute for Clinical Research, National Hospital Organization Osaka National Hospital, Chuo-ku, Osaka 540-0006, Japan; kanemura.yonehiro.hk@mail.hosp.go.jp; 7Department of Neurosurgery, Osaka International Cancer Institute, Chuo-ku, Osaka 541-8567, Japan

**Keywords:** glioblastoma, non-enhancing tumor (NET), ratio of T1- and T2-weighted images, MR relaxometry, ^11^C-methionine positron emission tomography

## Abstract

One of the challenges in glioblastoma (GBM) imaging is to visualize non-enhancing tumor (NET) lesions. The ratio of T1- and T2-weighted images (rT1/T2) is reported as a helpful imaging surrogate of microstructures of the brain. This research study investigated the possibility of using rT1/T2 as a surrogate for the T1- and T2-relaxation time of GBM to visualize NET effectively. The data of thirty-four histologically confirmed GBM patients whose T1-, T2- and contrast-enhanced T1-weighted MRI and ^11^C-methionine positron emission tomography (Met-PET) were available were collected for analysis. Two of them also underwent MR relaxometry with rT1/T2 reconstructed for all cases. Met-PET was used as ground truth with T2-FLAIR hyperintense lesion, with >1.5 in tumor-to-normal tissue ratio being NET. rT1/T2 values were compared with MR relaxometry and Met-PET. rT1/T2 values significantly correlated with both T1- and T2-relaxation times in a logarithmic manner (*p* < 0.05 for both cases). The distributions of rT1/T2 from Met-PET high and low T2-FLAIR hyperintense lesions were different and a novel metric named Likeliness of Methionine PET high (LMPH) deriving from rT1/T2 was statistically significant for detecting Met-PET high T2-FLAIR hyperintense lesions (mean AUC = 0.556 ± 0.117; *p* = 0.01). In conclusion, this research study supported the hypothesis that rT1/T2 could be a promising imaging marker for NET identification.

## 1. Introduction

One of the significant problems of MRI for glioblastoma (GBM) is its inability to visualize non-enhancing tumor (NET) lesions. This issue is clinically significant, as maximum tumor resection is considered one of the key prognostic factors for GBM treatment [1,2,3]. While conventional maximum tumor resection was based on contrast-enhancing lesions, it is becoming more apparent that NET within T2/FLAIR high-intensity lesions should also be set as a target for removal [2,4]. Tracer-based imaging such as ^11^C-methionine positron emission tomography (Met-PET) has been challenged for detecting NET and is considered the closest imaging modality to fulfill this goal [5,6,7,8,9,10,11]. However, due to ^11^C’s short half-life time, logistic and regulatory issues hamper Met-PET from becoming a gold-standard imaging modality worldwide. Thus, developing a novel MRI-based imaging method for NET visualization is necessary.

We recently reported that T1- and T2-relaxation time correlates with glioma tissues’ tumor cell density [12]. Furthermore, we demonstrated the possibility of using T1-relaxation time as an imaging surrogate for visualizing NET. However, we are also aware that T1- and T2-relaxometry is still not a clinically routine imaging sequence for brain tumors. On the other hand, the ratio of T1- and T2-weighted images (T1w/T2w-ratio map: rT1/T2) is reported as a helpful imaging surrogate of microstructures of the brain, which helps visualize multiple stenosis lesions [13,14,15,16]. This research study investigates the possibility of using rT1/T2 as a surrogate for the T1- and T2-relaxation time of GBM to visualize NET effectively.

## 2. Materials and Methods

### 2.1. Patient Selection

Thirty-four histologically confirmed GBM patients (15 women and 19 men, with a median age of 63.5 years old) at the Osaka International Cancer Institute and Osaka University Hospital whose T1-, T2- and contrast-enhanced T1-weighted MRI and MET-PET were available were included for this retrospective analysis. The pathological diagnosis was based on the 2016 World Health Organization classification for central nervous system tumors.

### 2.2. Image Co-Registration and Voxel-Of-Interest (VOI) Definition

Contrast-enhanced T1-weighted images (T1WIs) were co-registered to T2WIs. We used Vinci image-analyzing software (Max Planck Institute for Neurological Research Cologne; http://www.nf.mpg.de/vinci/, accessed on 4 January 2022) for image registration. Voxels-of-interest (VOIs) were defined by subtracting contrast-enhancing regions from the T2-FLAIR hyperintense lesion, as we only focused on NET within the T2-FLAIR hyperintense lesion for this research study (Figure 1). Investigational images such as Met-PET, T1- and T2-relaxation maps and rT1/T2, described in detail below, were also co-registered to T2WIs. The data obtained by a voxel-based analysis may be noisier than comparing histogram values such as maximum, minimum and average within the VOI, possibly rendering statistically significant findings insignificant. However, we accepted these disadvantages and focused our investigation on voxel-based analyses throughout the research study, as our final intention is to challenge visualizing NET in a voxel-based manner.

### 2.3. T1- and T2-Relaxometry by MP2RAGE and Multi-Echo T2WIs

Imaging was performed on a 3T MR scanner (Prisma; Siemens Healthcare, Erlangen, Germany). In total, 2 out of the 34 patients underwent T1- and T2-relaxometry. T1-relaxometry was achieved by first acquiring MP2RAGE images, then converting those images into T1-relaxation time maps. T2-relaxometry was achieved by first acquiring multi-echo T2-weighted images and then converting those images into T2-relaxation time maps, with both relaxometries performed via Bayesian inference modeling (Olea Nova+; Canon Medical Systems, Tochigi, Japan). MP2RAGE images were acquired using the following parameters: repetition time (TR) = 5000 ms; echo 112 time (TE) = 3.86 ms; and inversion time (TI) = 935/2820 ms. Multi-echo T2-weighted imaging was acquired using TR = 4000 ms and TE = 20, 40, 60, 80, 100, 120 and 140 ms. An additional 20 min scan was necessary on top of routine clinical imaging of GBM patients. Further details can be found for patients 8 and 11 in our previous publication [12].

### 2.4. ^11^C-Methionine Positron Emission Tomography (Met-PET)

PET studies were performed using an Eminence-G system (Shimadzu, Kyoto, Japan), with MET synthesized according to the method described by Hatakeyama et al. [17] and injected intravenously at a dose of 3 MBq/kg body weight. Tracer accumulation was recorded for 12 min in 59 or 99 transaxial sections over the entire brain. Summed activity from 20 to 32 min after tracer injection was used for image reconstruction. Images were stored in 256 × 256 × 59 or 99 anisotropic voxels, with each voxel being 1 × 1 × 2.6 mm. An area of high cell density was defined as those voxels presenting a tumor-to-normal tissue ratio (T/N ratio) >1.5. This cut-off was derived from previous publications showing that the T/N ratio = 1.5 was roughly equivalent to tissues with a cell density of 2000 cells/mm^2^. As cell density of healthy brain tissue ranges from 382 to 1106 cells/mm^2^, this cut-off was considered the most appropriate for defining with confidence those regions carrying a high tumor load [7,10,11,18].

### 2.5. T1w/T2w-Ratio Map (rT1/T2) Reconstruction

The T1w/T2w-ratio map (rT1/T2) was obtained by in-house imaging software incorporating the algorism developed by Ganzetti et al. [14]. The algorism and MATLAB codes for creating rT1/T2 can be found in “MRtool”, an open-source toolbox for SPM12 provided by Ganzetti et al. (https://www.nitrc.org/projects/mrtool/, accessed on 4 January 2022). Bias field correction was applied to the original T1WIs and T2WIs using SPM12 (https://www.fil.ion.ucl.ac.uk/spm/, accessed on 4 January 2022). Unbiased images were calibrated by adjusting the intensity histograms using the eye’s and temporal muscle’s lowest and highest intensity peaks. Thus, the intensity normalization of images aiming for trans-institutional image harmonization was not necessary for preprocessing in calculating the rT1/T2 map. The T2WI was co-registered to the T1WI, further producing rT1/T2 [16]. An example case presentation for rT1/T2 reconstruction is provided in Appendix A.

### 2.6. Analysis of the Relationship between T1- and T2-Relaxation Maps and rT1/T2

We first evaluated whether rT1/T2 could be a reliable surrogate for the T1- and T2-relaxation time. This analysis was conducted using the data from two GBM patients whose T1- and T2-relaxation maps were available. The rT1/T2 values within the above-mentioned VOIs were plotted as a function of T1- and T2-relaxation time, thus enabling quantitative comparison of these modalities in a voxel-wise manner.

### 2.7. Comparing rT1/T2 and Met-PET and Defining the “Likeliness of Met-PET High”

Next, we analyzed the relationship between the rT1/T2 values and Met-PET, as Met-PET is the closest imaging modality to estimate tumor cell density for gliomas. The VOI was segmented into either “Met-PET high” or “Met-PET low” at a threshold of 1.5 in tumor-to-normal tissue ratio (T/N ratio). The histograms of rT1/T2 from “Met-PET high” or “Met-PET low” VOIs were plotted and these data were further used to calculate the “Likeliness of Met-PET high (LMPH)”, an index which reflects the likeliness of Met-PET being higher than 1.5 in T/N ratio, as previously described. The “LMPH” at a given bin *k* was defined using the following equation:(1)$$\mathrm{LMPH}\;(k)=\left\{\frac{n_{H}\left(k\right)}{N_{H}\;}-\frac{\;n_{L}\left(k\right)}{N_{L}}\right\}/\left\{\frac{n_{H}\left(k\right)}{N_{H}}+\frac{\;n_{L}\left(k\right)}{N_{L}}\right\}\;,$$
where *k* denotes a given bin of the histogram, $n_{H}\left(k\right)$ is the number of Met-PET high voxels for bin *k*, $n_{L}\left(k\right)$ is the Met-PET low voxel count for bin *k*, $N_{H}$ is the total number of high ^11^C-methionine uptake voxels and $N_{L}$ is the total number of low ^11^C-methionine uptake voxels. When LMPH was uncalculatable because both $n_{H}\left(k\right)$ and $n_{L}\left(k\right)$ were 0, the corresponding rT1/T2 voxels were removed from the analysis.

Once the correlation between rT1/T2 and LMPH was obtained, rT1/T2 could be converted into an LMPH map that reflected the likeliness of Met-PET high in a voxel-wise manner. The MATLAB code is provided in the Appendix A (rT1T2tolikeliness.m).

### 2.8. Prediction Accuracy Estimation Analysis of LMPH Deriving from rT1/T2

To evaluate whether the LMPH deriving from rT1/T2 helped predict Met-PET high and low lesions in T2/FLAIR high-intensity lesions of GBM, we tested the prediction accuracy with leave-one-patient-out cross-validation. More specifically, we calculated the LMPH from rT1/T2 using data from 33 patients leaving out one for validation. Next, we applied the obtained rT1/T2 to the LMPH conversion matrix to the left-out patient in a voxel-wise manner. The rT1/T2 of the validation patient was converted to an LMPH map and we measured the accuracy for predicting Met-PET high and low via LMPH. This procedure was repeated for all patients except one, whose $N_{H}$ was 0. The overall prediction accuracy is reported as the mean of the area under the receiver operating characteristic (ROC) curve (AUC) computed for each patient.

### 2.9. Statistical Analysis

The Pearson correlation coefficient was calculated between rT1/T2 and T1- and T2-relaxation time. A one-sample *t*-test was used to test the classification accuracy of the LMPH for classifying Met-PET high and low lesions.

## 3. Results

### 3.1. rT1/T2 Logarithmically Correlated with T1- and T2-Relaxation Time

Seventy-six thousand seven hundred six voxels from two patients were analyzed. As shown in Figure 2, the rT1/T2 values significantly correlated with both T1- and T2-relaxation times in a logarithmic manner. The correlations between rT1/T2 and T1- and T2-relaxation times were rT1/T2 = 0.862 × e^−0.0000213 × T1-relax^ and rT1/T2 = 0.951 × e^−0.000884 × T2-relax^ (r = −0.043 and −0.259; *p* < 0.05).

### 3.2. rT1/T2 Distributions of Met-PET High and Low T2-FLAIR Hyperintense Lesions

Figure 3A demonstrates that the rT1/T2 values mainly distributed within the range from 0.3 to 1.3 and that the distributions of rT1/T2 from Met-PET high and low T2-FLAIR hyperintense lesions were different. The probability distribution of rT1/T2 from Met-PET high lesions was narrower than from Met-PET low T2-FLAIR hyperintense lesions. Thus, these data supported further attempts to separate Met-PET high and low T2-FLAIR hyperintense lesions via rT1/T2.

### 3.3. Likeliness of Met-PET High Derived from rT1/T2 (LMPH) Helped Classify Met-PET High and Low T2-FLAIR Hyperintense Lesions

The AUCs ranged from 0.323 to 0.798 for the 33 patients (Appendix A). The mean AUC obtained from classifying Met-PET high and low T2-FLAIR hyperintense lesions by LMPH was significantly higher than 0.5 (chance accuracy) (mean AUC = 0.556 ± 0.117; *p* = 0.01) (Figure 3B).

### 3.4. LMPH Map for Visualizing Met-PET High NET

The LMPH values are plotted as a function of rT1/T2 using all available 34 cases in Figure 4A, the correlation table is provided in Appendix A and the conversion matrix as a MATLAB file is provided in the Appendix A (B20211211_Likeliness_All_20211219.mat). An LMPH map could be reconstructed from rT1/T2 and two representative cases are shown (Figure 4B,C). In both presented cases, T2-FLAIR hyperintense lesions expanded anteriorly and posteriorly to the contrast-enhancing lesion. Conventional contrast-enhanced T1WIs and T2WIs could not distinguish whether the T2-FLAIR hyperintense lesions were merely due to vasogenic edema or NET. On the other hand, Met-PET clearly showed that anterior T2-FLAIR hyperintense lesions were more tumorous than posterior lesions. The reconstructed LMPH map helped distinguish NET and vasogenic T2-FLAIR hyperintense lesions.

## 4. Discussion

Identifying NET is a crucial yet challenging issue in glioma patient care [19]. Contrast-enhancing lesions have long been the main target of resection and irradiation for GBM patient care, as it has been thought that these lesions represent highly tumorous tissues [3]. This concept is justified to some extent, because the amount of contrast-enhancing lesion resections positively correlates with patient prognosis for GBM [3]. On the other hand, the research community has also known for a long time that non-enhancing lesions could also contain tissues with extremely high tumor cell density, indistinguishable from those in contrast-enhancing lesions. Our stereotactic tissue sampling study showed that the difference in tumor cell densities between contrast-enhancing and non-enhancing lesions was insignificant [18]. Furthermore, several reports demonstrated that the prognosis of GBM patients could extend even more if one could resect T2-FLAIR hyperintense lesions beyond contrast-enhancing lesions [2,4]. These findings highlight the clinical necessity of distinguishing NET from vasogenic edema within T2-FLAIR hyperintense lesions to accurately evaluate the extension of highly tumorous tissues within the brain.

Amino acid tracer-based positron emission tomography (PET) is one of the imaging modalities that has thoroughly been investigated for NET visualization. Notably, several stereotactic tissue sampling studies demonstrated that Met-PET well correlates with glioma cell density [5,11,12,18] and that its diagnostic accuracy was higher than that of conventional MRI [20]. However, incorporating amino acid tracer-based PET into daily clinical practice has still not been achieved and is still considered an investigational imaging modality. On the other hand, MRI is an exceptionally well-adopted imaging modality. The research community has striven to develop novel technologies such as MR spectroscopy and chemical exchange saturation transfer (CEST) imaging that allow us to recognize NET [21,22], which are in the process of clinical validation. In line with this research trend, our group investigated the possibility of using MR relaxometry to delineate NET. Our previous finding suggested that MR relaxometry, especially T1-relaxation time, helped to identify NET [12]. As MR relaxometry requires an additional 20 min scan and is still not a routine clinical imaging technique, this research study tested the hypothesis that the ratio of T1- and T2-weighted images (T1w/T2w-ratio map: rT1/T2) could be an imaging surrogate of MR relaxometry and could further be used for visualizing NET. The scientific rationale of pursuing rT1/T2 as an MR relaxometry surrogate in GBM patients is based on previous observations that rT1/T2 was somewhat correlated with the tissue microstructure of multiple sclerosis. Although the cause of abnormal observations of rT1/T2 in multiple sclerosis is still under debate, the proposal of using rT1/T2 as MR relaxometry is highly appealing, as rT1/T2 can be reconstructed from conventional T1WIs and T2WIs [16]. Our findings in the current research study support the initial hypothesis that rT1/T2 could be a promising imaging marker for NET identification and we were able to demonstrate that the LMPH map deriving from rT1/T2 could help identify NET (Figure 4B,C).

This study has several limitations and issues to be discussed. First, this study was conducted using data from only two domestic institutions, which might reduce the generalizability of the results. Furthermore, the sample size was limited to 34, which hampers drawing a definite conclusion. Validation studies supported by a more extensive sample size collecting data from various institutions is necessary. Second, the current study used Met-PET as the fundamental ground truth that reflects NET. However, ideally, histopathological data obtained by careful tissue sampling should be used as a reference. Finally, the authors are aware that classification accuracy was not high. The classification performance of LMPH (mean AUC = 0.556) is insufficient for clinical use. A simple implementation of this method into routine pre- or intra-operative surgical planning seems challenging for detecting NET. However, combining rT1/T2 and other MR-based surrogates such as CEST and MR spectroscopy could be a promising research avenue to pursue.

## 5. Conclusions

The rT1/T2 values significantly correlated with both T1- and T2-relaxation times in a logarithmic manner. The distributions of rT1/T2 from Met-PET high and low T2-FLAIR hyperintense lesions were different and the LMPH deriving from rT1/T2 was statistically significant for detecting Met-PET high T2-FLAIR hyperintense lesions.

## Figures and Tables

**Figure 1 brainsci-12-00099-f001:**
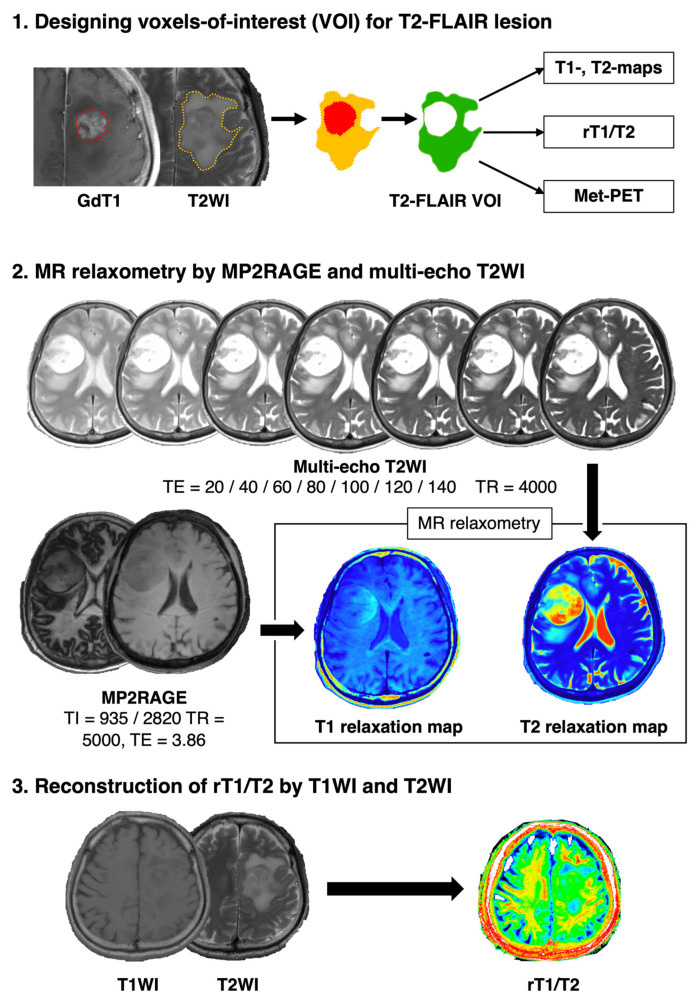
The analytical scheme of this study is presented. VOIs were depicted by subtracting the contrast-enhancing region from the T2-FLAIR hyperintense lesion. These VOIs were applied to the images of interest, such as T1- and T2-relaxation maps, rT1/T2 and Met-PET (1). Bayesian inference modeling was utilized to convert MP2RAGE images and multi-echo T2WIs to T1- and T2-relaxation maps (2). The rT1/T2 image was reconstructed by image co-registration and intensity correction as mentioned below (3).

**Figure 2 brainsci-12-00099-f002:**
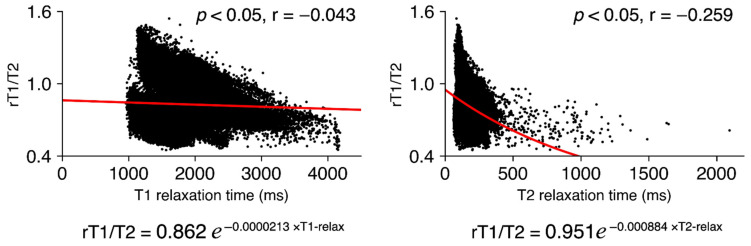
The logarithmic correlations between rT1/T2 and T1- and T2-relaxation times are presented.

**Figure 3 brainsci-12-00099-f003:**
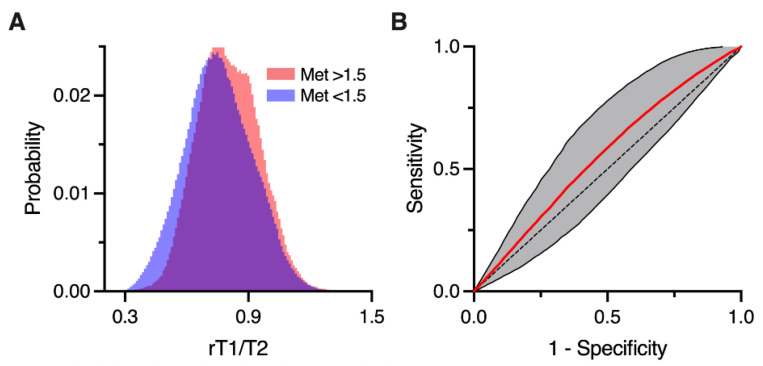
The probability distributions of rT1/T2 corresponding to high Met-PET (T/N ratio > 1.5) or low Met-PET (T/N ratio < 1.5) within T2-FLAIR hyperintense lesions are presented (**A**). Note that the probability distribution of rT1/T2 corresponding to high Met-PET was narrower than low Met-PET. Classification accuracy of LMPH for classifying Met-PET high or low T2-FLAIR hyperintense lesions is shown (**B**). The red line indicates the mean receiver operating characteristic curve and the gray shaded area denotes the standard deviation of the curve.

**Figure 4 brainsci-12-00099-f004:**
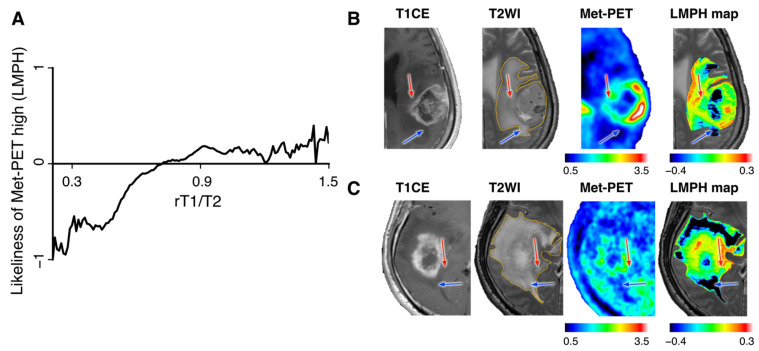
LMPH values are plotted as a function of rT1/T2 using all available 34 cases (**A**). Representative cases demonstrating the usefulness of the LMPH map are shown (**B**,**C**). Red and blue arrows indicate the lesions whose Met-PET high or low was accurately predicted by LMPH within T2/FLAIR high-intensity lesions, respectively.

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
