# Peer review of "Prediction and Visualization of Non-Enhancing Tumor in Glioblastoma via T1w/T2w-Ratio Map"

_brainsci, 2022, doi:10.3390/brainsci12010099_

Round 1
Reviewer 1 Report
In the present work, the authors investigate the use of the T1w / T2w ratio to identify NET in glioblastomas.
As a fundamental truth, they use the findings of Met-PET and also correlate the findings of the T1 / T2 ratio with MR relaxometry, as it is a substitute for this modality.
Strengths:
- The introduction is very well structured.
- The methodology is rigorous and the results are clearly presented.
- The hypothesis has great clinical relevance, since distinguishing edema from non-enhancing tumor is an objective that has been sought for several years.
Major concerns:
- The sample size is small to be able to draw conclusions.
- The results of the validation are insufficient to generalize the results.
- The pre-processing of the images should be improved in order to facilitate the harmonization of images that come from different scanner manufacturers.
- The voxel-based analysis adds complexity to the handling of the data, the noise of the extracted data can hinder its statistical treatment.
- The use of Met-PET as a fundamental truth is highly debatable, I would reserve that term for histopathological analysis based on samples.
Minor concerns:
- The process of preparing the T1w / T2w ratio maps must be explained in greater detail. Offering the code freely accessible would be an important step for other institutions to validate these results.
Author Response
# Reviewer 1
Critique 1: The sample size is small to be able to draw conclusions and the results of the validation are insufficient to generalize the results.
Response: The authors deeply agree with the comment mentioned above raised by the reviewer. We agree that further validation study is crucial to support our finding. We now added the following statement in line 281 as follows.
“Furthermore, the sample size was limited to 34, which hampers drawing a definite conclusion. Validation studies supported by a more extensive sample size collecting data from various institutions is necessary.”
Critique 2: The pre-processing of the images should be improved in order to facilitate the harmonization of images that come from different scanner manufacturers.
Response: The algorithm developed by Ganzetti et al. incorporates image pre-processing within their package. Thus, the inputs for reconstructing rT1/T2 did not require pre-processing, and raw T1WI and T2WI were sufficient. We now provide an example during image processing in Figure S1. We also now added the following statement in line 124 as follows.
“Bias field correction was applied to the original T1WI and T2WI using SPM12 (https://www.fil.ion.ucl.ac.uk/spm/). Unbiased images were calibrated by adjusting the intensity histograms using the eye and temporal muscle's lowest and highest intensity peaks. Thus, intensity normalization of images aiming for trans-institutional image harmonization was not necessary for preprocessing in calculating rT1/T2 map. T2WI was co-registered on T1WI, further producing rT1/T2 [16]. An example case presentation for rT1/T2 reconstruction is provided in Figure S1.”
Critique 3: The voxel-based analysis adds complexity to the handling of the data, the noise of the extracted data can hinder its statistical treatment.
Response: We fully agree with the raised concern. However, our final challenge was to develop a method to visualize NET. Arbitrarily lowering the resolution of the data or using a histogram-based analysis within the VOI was not thought appropriate for this aim. Thus, we analyzed the data with full resolution throughout the research. We explain our intention in the revised manuscript in line 81 as follows.
“Obtained data by a voxel-based analysis may be noisier than comparing histogram values such as maximum, minimum, and average within the VOI, possibly rendering statistically significant findings insignificant. However, we accepted these disadvantages and focused our investigation on voxel-based analyses throughout the research, as our final intention was to challenge visualizing NET in a voxel-based manner.
Critique 4: The use of Met-PET as a fundamental truth is highly debatable, I would reserve that term for histopathological analysis based on samples.
Response: We agree with the raised concern. Although our group still believes that Met-PET can be considered “the ground truth” for visualizing high-grade glioma infiltration into the brain, we also understand that this top is still in great debate. We now added the following statement in line 283 as follows.
“Second, the current study used Met-PET as the fundamental ground truth that reflects NET. However, ideally, histopathological data obtained by careful tissue sampling should be used as a reference.
Critique 5: The process of preparing the T1w / T2w ratio maps must be explained in greater detail. Offering the code freely accessible would be an important step for other institutions to validate these results.
Response: We agree with the reviewer that providing details for reconstructing T1w/T2w is critical for the current research. As we have already documented in the original version of the manuscript, the algorithm developed by Ganzetti et al. was incorporated into our image analyzing pipeline. We now provide an example during image processing in Figure S1. Furthermore, we now provide our original code in Matlab, which converts rT1/T2 to “Likeliness of Met-PET high” map. We also now added the following statement in line 117 as follows.
“T1w/T2w-ratio map (rT1/T2) was obtained by in-house imaging software incorpo-rating the algorism developed by Ganzetti et al. [14]. The algorism and Matlab codes for creating rT1/T2 can be found in “MRtool”, an open-source toolbox for SPM12 provided by Ganzetti et al. (https://www.nitrc.org/projects/mrtool/, accessed on January 4, 2022). Bias field correction was applied to the original T1WI and T2WI using SPM12 (https://www.fil.ion.ucl.ac.uk/spm/). Unbiased images were calibrated by adjusting the intensity histograms using the eye and temporal muscle's lowest and highest intensity peaks. Thus, intensity normalization of images aiming for trans-institutional image harmonization was not necessary for preprocessing in calculating rT1/T2 map. T2WI was co-registered on T1WI, further producing rT1/T2 [16]. An example case prestation for rT1/T2 reconstruction is provided in Figure S1.
Reviewer 2 Report
This is a nice study, dealing with an important issue. It would be a great help, if there would be a possibility to recognise real tumor spread from oedema. I do not consider it for fundamental for the publication but one of question is, if this method could be implemented in preoperative planning or even in intraoperative MRI protocol. Based on the images in the publication, it would be a little bit difficult to use the images provided for this issue. Do you have an example image that could help to improve surgical planning or is there further project for this part.
Author Response
# Reviewer 2
Critique 1: This is a nice study, dealing with an important issue. It would be a great help, if there would be a possibility to recognise real tumor spread from oedema. I do not consider it for fundamental for the publication but one of question is, if this method could be implemented in preoperative planning or even in intraoperative MRI protocol. Based on the images in the publication, it would be a little bit difficult to use the images provided for this issue. Do you have an example image that could help to improve surgical planning or is there further project for this part.
Response: The authors thank very kind comments and reviews. We agree with the reviewer that incorporating this technique into routine pre- or intra-operative surgical planning is the goal of the study. Although mere implementation is technically not difficult, the authors are also aware that our findings should be critically evaluated before being used for real-world patient care. We added the following comments in line 287 in the revised manuscript.
“A simple implementation of this method into routine pre- or intra-operative surgical planning seems challenging for detecting NET. However, combining rT1/T2 and other MR-based surrogates such as CEST and MR spectroscopy could be a promising research avenue to pursue.”
Round 2
Reviewer 1 Report
The authors have satisfactorily complied with the requirements of the review.